# A novel mechanism of streptomycin resistance in *Yersinia pestis*: Mutation in the *rpsL* gene

**Ruixia Dai**[1,2◉], **Jian He**[1,2◉], **Xi Zha**[3◉], **Yiting Wang**[4,5◉], **Xuefei Zhang**[1,2◉], **He Gao**[4,5], **Xiaoyan Yang**[1,2], **Juan Li**[4,5], **Youquan Xin**[1,2], **Yumeng Wang**[4,5], **Sheng Li**[1,2], **Juan Jin**[1,2], **Qi Zhang**[1,2], **Jixiang Bai**[1,2], **Yao Peng**[4,5], **Hailian Wu**[1,2], **Qingwen Zhang**[1,2], **Baiqing Wei**[1,2], **Jianguo Xu**[4,5], **Wei Li**[4,5]*

1 Qinghai Institute for Endemic Disease Control and Prevention, Xining, China, 2 Key Laboratory of the National Health Commission for Plague Control and Prevention, Xining, China, 3 Center for Disease Control and Prevention of Tibet Autonomous Region, Lhasa, China, 4 National Institute for Communicable Disease Control and Prevention, China CDC, Changping, Beijing, China, 5 State Key Laboratory of Infectious Disease Prevention and Control, Beijing, China

◉ These authors contributed equally to this work.
* liwei@icdc.cn

**Data Availability Statement:** The sequencing data of Y. pestis strain S19960127 are available in GenBank under accession numbers CP045636–CP045640, and the genome sequences of Y. pestis

## Abstract

Streptomycin is considered to be one of the effective antibiotics for the treatment of plague. In order to investigate the streptomycin resistance of *Y. pestis* in China, we evaluated streptomycin susceptibility of 536 *Y. pestis* strains in China *in vitro* using the minimal inhibitory concentration (MIC) and screened streptomycin resistance-associated genes (*strA* and *strB*) by PCR method. A clinical *Y. pestis* isolate (S19960127) exhibited high-level resistance to streptomycin (the MIC was 4,096 mg/L). The strain (biovar antiqua) was isolated from a pneumonic plague outbreak in 1996 in Tibet Autonomous Region, China, belonging to the *Marmota himalayana* Qinghai–Tibet Plateau plague focus. In contrast to previously reported streptomycin resistance mediated by conjugative plasmids, the genome sequencing and allelic replacement experiments demonstrated that an *rpsL* gene (ribosomal protein S12) mutation with substitution of amino-acid 43 (K43R) was responsible for the high-level resistance to streptomycin in strain S19960127, which is consistent with the mutation reported in some streptomycin-resistant *Mycobacterium tuberculosis* strains. Streptomycin is used as the first-line treatment against plague in many countries. The emergence of streptomycin resistance in *Y. pestis* represents a critical public health problem. So streptomycin susceptibility monitoring of *Y. pestis* isolates should not only include plasmid-mediated resistance but also include the ribosomal protein S12 gene (*rpsL*) mutation, especially when treatment failure is suspected due to antibiotic resistance.

## Author summary

The plague natural foci are widely distributed in the world, and correspondingly, the plague still poses a significant threat to human health in some countries with endemic

strains sequenced in this study have been deposited in GenBank with accession numbers WHKG00000000–WHLN00000000.

**Funding:** RD received support from National Natural Science Foundation of China (81660349), WL from National Important Scientific and Technology Project (2018ZX10101002-002), RD from Science and Technology Plan Project in Qinghai Province (2019-ZJ-7074) and XZ from National Health Commission Project for Key Laboratory of Plague Prevention and Control (2019PT310004). The funders had no role in study design, data collection and analysis, decision to publish, or preparation of the manuscript.

**Competing interests:** The authors have declared that no competing interests exist.

plague foci. Streptomycin is used as the first-line treatment against plague in many countries for the antibiotic is considered to be one of the effective antibiotics, particularly for the treatment of pneumonic plague. The resistance to streptomycin had been reported in *Y. pestis* strains from Madagascar in previous studies. In this study, we reported the high-level resistance to streptomycin in a clinical isolate of *Y. pestis* from a pneumonic patient in Tibet Autonomous Region, China, and a novel mechanism of streptomycin resistance, i.e. mutation in the *rpsL* gene were identified. The knowledge acquired about streptomycin resistance in *Y. pestis* will remain of great practical value. For the emergence of resistance to streptomycin in *Y. pestis* would render the treatment failure, thus corresponding antibiotic monitoring should be routinely carried out in countries threatened by plague. In addition, based on our further understanding about streptomycin resistance of *Y. pestis* isolates, such monitoring should not only include plasmid-mediated resistance but also include the ribosomal protein S12 gene (*rpsL*) mutation in *Y. pestis* isolates.

## Background

Plague is an acute infectious disease caused by *Yersinia pestis*; it is primarily a disease of wild rodents and their parasitic fleas are considered to be the transmitting vectors. Three major types of plague occur in human beings: bubonic, pneumonic, and septicemic plague. Pneumonic plague is the most threatening clinical form as person-to-person transmission typically occurs. So far, five biovars of *Y. pestis* have been recognized on the basis of their biochemical properties: *Y. pestis* antiqua, mediaevalis, orientalis, pestoides (microtus), and intermedium [1–3].

Streptomycin is the most effective antibiotic agent against *Y. pestis* [4]. However, resistance to streptomycin has been reported in *Y. pestis* strains from Madagascar in two studies [5,6]. In 1995, an isolate named 17/95 was isolated from a 16-year-old boy in the Ambalavao district of Madagascar and this strain exhibited multidrug-resistant traits to eight antimicrobial agents (streptomycin, chloramphenicol, tetracycline, sulfonamides, ampicillin, kanamycin, spectinomycin, and minocycline) [5]. The biovar of the strain 17/95 was orientalis and it carried the conjugative multidrug-resistant plasmid pIP1202 (a member of the IncA/C plasmid family), in which the high level of streptomycin resistance (MICs>2,048 mg/L) was due to the presence of streptomycin phosphotransferase activity induced by streptomycin resistance-associated genes (*strA* and *strB*) [5]. Another isolate named 16/95 (orientalis), only exhibiting streptomycin resistance, was obtained in 1995 in the Ampitana district of Madagascar from an axillary bubo puncture from a 14-year-old boy before antibiotic treatment [6]. The resistance determinant of 16/95 was carried by a self-transferable plasmid (pIP1203, a member of the IncP group) and the MIC of streptomycin for 16/95 was 1,024 mg/L. Recently, a paper reported plasmid-mediated doxycycline resistance in a *Y. pestis* strain in Madagascar (isolated from a rat in 1998) [7]. In addition, another multidrug-resistant *Y. pestis* strain was isolated in Mongolia in 2000 from a marmot, but the genetic basis and transferability of the resistance were not investigated [7,8].

Public health measures and effective antibiotic treatments led to a drastic decrease in plague worldwide. However, the disease has not yet been eradicated, and endemic plague foci are widely distributed in Africa, Asia, and North and South America [9]. In this study, we report high-level resistance to streptomycin in a clinical isolate of *Y. pestis* named S19960127 from a pneumonic patient occurred in 1996 in Shannan Prefecture, Tibet Autonomous Region, China. Different from the mechanism of conjugative plasmids that carry

phosphoryltransferase encoded by *strA* or *strB* to inactivate streptomycin[5,6], we found that the high-level streptomycin resistance originated from a mutation in *rpsL* (ribosomal protein S12).

## Methods

### Ethics statement

The ethical aspect of this study was approved by the Qinghai Institute for Endemic Disease Control and Prevention, Xining, China. All procedures were in accordance with the ethical standards of the National Research Committee.

### Antibiotic resistance evaluation

A total of 536 *Y. pestis* isolates from 1946 to 2012 in 12 natural plague foci in China were used to evaluate the susceptibility to streptomycin. Susceptibility testing for *Y. pestis* and corresponding quality control referenced standard Clinical and Laboratory Standards Institute (CLSI) methods [10]. Minimal inhibitory concentrations (MICs) were determined by the agar dilution method following National Committee for Clinical Laboratory Standards guidelines [11] and previous literatures [12,13]. Quality control strains (*Escherichia coli* ATCC 25922 and *Pseudomonas aeruginosa* ATCC27853) were tested periodically with each batch of *Y. pestis* isolates to validate the accuracy of the procedure. All the experiments associated with *Y. pestis* were conducted in the Bio-safety Level 3 (BSL-3) laboratory in Qinghai Institute for Endemic Disease Control and Prevention.

### PCR for screening the streptomycin resistance-associated genes *strA* and *strB*

We scanned for the streptomycin resistance-associated genes (*strA* and *strB*) using the PCR method in 536 *Y. pestis* strains, in which the *strA* and *strB* genes targeted the enzymes that phosphorylated streptomycin in the conjugative plasmids pIP1202 (*Y. pestis* strain 17/95) and pIP1203 (*Y. pestis* strain 16/95) (EMBL data bank under accession number AJ249779) [5,6]. The primers used to amplify *strA* and *strB* are listed in S1 Table. PCR was performed using Taq DNA polymerase (Takara) with the following cycling protocol: denaturing step for 5 min at 95˚C, followed by 30 cycles of the amplification step at 95˚C for 50 s, 58˚C for 50 s, and 72˚C for 1 min, and the final extension step for 5 min at 72˚C.

### *Y. pestis* and DNA preparation for genome sequencing

A total of 15 *Y. pestis* isolates from Shannan Prefecture (natural plague focus) in Tibet Autonomous Region were used for genome sequencing, including seven *Y. pestis* strains isolated from human and eight strains isolated from *Marmota himalayana* or hare in various years (S2 Table). The 15 strains were also included in the 536 *Y. pestis* isolates used for evaluation of antibiotic resistance. Genomic DNA from each bacterium was extracted using the QIAamp DNA Mini Kit (Qiagen Shanghai, China) according to the manufacturer's instructions. The complete genome of streptomycin resistance *Y. pestis* strain S19960127 was sequenced using PacBio RS II sequencers and assembled *de novo* using SMRT Link v5.1.0 software [14]. The draft genomes of the other 14 strains were sequenced on the Illumina HiSeq 2500-PE125 platform with massively parallel sequencing Illumina technology and assembled using SOAP software [15]. The PlasmidFinder was used to identify the characteristics of plasmids in sequenced strains [16].

## Identification of mutated genes in *Y. pestis* strain S19960127 and alignment of *rpsL* genes

The genes in the complete genome of the streptomycin-resistant *Y. pestis* strain S19960127 was predicted by Glimmer software using the default parameters [17]. Compared with the streptomycin-sensitive *Y. pestis* strains in the Genbank database, those genes with point mutations in S19960127 were blasted and an *rps*L gene (ribosomal protein S12) mutation was finally found. This SNP point mutation was confirmed by PCR (primers: rpsL-F/rpsL-R in S2 Table) and Sanger sequencing. MEGA6.0 software [18] was used to align the mutated *rpsL* gene in *Y. pestis* S19960127 with its counterparts in *M. tuberculosis* [19], *E. coli*, and *Salmonella*. Another gene *rrs*, also responsible for streptomycin-resistance in *M. tuberculosis* [20], was examined in S19960127.

## Gene replacement of *rpsL*

Allele exchange was carried out as described previously [21,22]. The primers used in this study are listed in S1 Table. The chloramphenicol acetyltransferase (cat) gene was amplified from the pKD3 plasmid with the primers Dcat-F/Dcat-R. The tool plasmids were constructed as follows. The primers YPO0199-DF1/YPO0199-DR1 and YPO0199-DF2/YPO0199-DR2 were used to amplify 507 bp upstream and 495 bp downstream of the *rpsL* gene from *Y. pestis* EV76 genomic DNA (note: EV76 is a vaccine strain and streptomycin sensitive, MIC: 4mg/L). The overlapping PCR fragment was cleaved with *XhoI* and *SpeI* enzymes then ligated to the suicide plasmid pWM91 digested with the same enzymes. In order to delete the locus of *rpsL* gene in S19960127, the recombinant plasmid (pWM△YPO0199::*cat*) was transferred from SM10λpir to S19960127 by conjugation, and selection of transformants were performed on ampicillin (100 μg/ml) plates and chloramphenicol (20μg/ml) plates overnight at 37˚C. In order to re-introduce the locus of *rpsL* gene with *rpsL*-128A, the primers YPO0199-DF1/YPO0199-DR2 were used to amplify a 1281-bp region that contained the *rpsL* gene from *Y. pestis* strain EV76 genomic DNA. The PCR fragment was ligated to the Pwm91 plasmid. The recombinant plasmids were transformed into *E. coli* SM10λpir and then combined with S19960127△YPO0199::*cat*. The mixtures were transferred to gentian violet agar plates at 37˚C to select *Y. pestis*. The *rpsL* mutant (*Y. pestis* S19960127:: *rpsL*-128A) was transferred to 10% sucrose (W/V) plates at 22˚C for 36–48 h to remove the plasmid backbone Pwm91.The replacement of the *rpsL* gene with the mutant allele (G128-A) was confirmed by PCR (primers: rpsL-F/rpsL-R) and DNA sequencing. Meanwhile the *caf1* gene was screened by PCR to guarantee the target bacterium was *Y. pestis*. In addition, the streptomycin resistance was evaluated by MIC method. In order to intuitively display the diminishment of streptomycin resistance by allele gene mutation, the disk diffusion method were used to illustrate the streptomycin resistance according to reference protocols [23].

## Results

### Streptomycin Resistance in *Y. pestis*

A clinical isolate of *Y. pestis* named S19960127 was highly resistant to streptomycin and the MIC was 4,096 mg/L, while the MIC breakpoint of streptomycin resistance stipulated by CLSI is ≥16 mg/L. Apart from *Y. pestis* strain S19960127, all the other 535 strains in this study were susceptible to streptomycin, and the ranges of MICs were 2–4 mg/L with the 50% and 90% MIC values all were 4 mg/L. Moreover, the streptomycin resistance genes *strA* and *strB* in the 536 strains were screened using PCR assays. None of them, including S19960127, carried these genes. Such results suggested that the high level of streptomycin resistance exhibited in strain

S19960127 is not due to the presence of conjugative plasmids or self-transferable plasmids like the isolates from Madagascar [5], [6].

## Pneumonic plague outbreak associated with streptomycin-resistant strain S19960127 in 1996 in Tibet, China

The strain S19960127 isolated from a pneumonic patient's organs at necropsy during a plague outbreak that occurred in 1996 in Qayü village (Latitude: 28.17; Logitude: 92.44), Lhünze County, Shannan Prefecture in Tibet, China. On 2 August 1996, a 21-year-old herdsman (Patient A) caught and skinned a diseased hare. On 4 August, Patient A suffered a fever with left axillary lymphadenopathy (bubonic plague) and the next day had the onset of headache, shivering, chest pain, and a productive cough with bloody sputum, and the patient died on 9 August without any treatment. On 11August, Patient B (a veterinarian) who once took care of Patient A without any personal protection became ill and showed corresponding symptoms with primary pneumonic plague. Patient B was given unified combination treatment intramuscular injection streptomycin (1 g, twice daily), oral trimethoprim/sulfamethoxazole (1 g, three times daily); oral tetracycline (0.5 g, three times daily), Such treatment lasted eight days and the patient died on 19 August. Patient B's wife (Patient C) suffered primary pneumonic plague on 16 August. Even although she was administrated above combination treatment, she died on 19 August. A *Y. pestis* strain (S19960127) was isolated from necropsy organ samples from Patient C, and the strain was identified with streptomycin resistance in this study. A village doctor (Patient D) who close contacted with Patients C during the course of her illness suffered fever and coughing with blood-tinged sputum on 17 August. Patient D was given same treatment on 17 August but still died on 22 August.

## General feature of streptomycin resistance strain S19960127

The whole genome of *Y. pestis* S19960127 (sequence accession number CP045636) consists of a 4.63-megabase chromosome and four plasmids, besides three common virulence plasmid pCD1/pYV, pPCP1/pPst, pMT1/ Fra [24,25], a novel 33.9-kb plasmid (named pS96127) in the *Y. pestis* S19960127 genome was discovered. The novel plasmid pS96127 shared 99.95% identity and 99.98 coverage with the pTP33 plasmid in *Y. pestis* strain I-2638 (KT020860.1). Apart from S19960127, another 14 strains sensitive to streptomycin isolated in Shannan Prefecture also carried the pS96127 plasmid.

## Mutation in the *rpsL* gene are involved in high-level streptomycin resistance in S19960127

The whole genome of *Y. pestis* strain S19960127 was analyzed to find the genetic differences for streptomycin resistance. In these alignment results, the *rpsL* gene that encodes ribosomal S12 protein was mutated at 128 bp in S19960127, corresponding to amino-acid substitution of Lys to Arg at site 43 (K43R) in the RpsL protein (Fig 1). Such streptomycin-resistant mutation is consistent with the ribosomal S12 mutation in *M. tuberculosis* [20,26]. However, those *Y. pestis* strains sensitive to streptomycin in this study, including *Y. pestis* S19960038 and S19960156 (isolated from a county adjacent to Lhünze County in 1996), as well as other strains in Shannan Prefecture, did not exhibit any mutation in the *rpsL* gene. In addition, no amino-acid alteration at position 88 of ribosomal protein S12 or *rrs* gene mutations in S19960127 were found (Figs 1 and S1), which also confer streptomycin resistance in *M. tuberculosis* [26].

```
                            41                                                          90
S.Typhimurium LT2  ...TPKKPNSALR KVCRVRLTNG FEVTSYIGGE GHNLQEHSVI LIRGGRVKDLP...
                      N                                                            S

E. coli            ...TPKKPNSALR KVCRVRLTNG FEVTSYIGGE GHNLQEHSVI LIRGGRVKDLP...

M. tuberculosis    ...TPKKPNSALR KVARVKLTSQ VEVTAYIPGE GHNLQEHSMV LVRGGRVKDLP...
                      R                                                            R
                      T                                                            Q

Y.pestis CO92      ...TPKKPNSALR KVCRVRLTNG FEVTSYIGGE GHNLQEHSVI LIRGGRVKDLP...

Y.pestis S19960038...TPKKPNSALR KVCRVRLTNG FEVTSYIGGE GHNLQEHSVI LIRGGRVKDLP...

Y.pestis S19960127    R
```

**Fig 1. Sequence comparisons of RpsL protein.** The relevant streptomycin resistance determinants in S19960127 compared with CO92 and a local susceptible isolate S19960038, as well as *M. tuberculosis*, *E.coli*, and *S.Typhimurium*. Each singly conferring streptomycin resistance are listed below the relevant positions. The numbering is based on reference sequences of the *E. coli* K-12 substrain MG1655 RpsL (accession number NP_417801). Amino acid substitutions in RpsL were observed at positions 43 in *Y.pestis* S19960127.

### Allelic gene replacement experiments demonstrated the mechanism that the *rpsL* mutation at 128 bp is the cause of streptomycin resistance in S19960127

To confirm the role of the observed mutations in streptomycin resistance, allelic exchanges experiments were carried out in S19960127, leading to the replacement of *rpsL-128G* in S19960127 with *rpsL-128A* from *Y.pestis* strain EV76 (MIC of streptomycin: 4 mg/L). As indicated in Fig 2, such genetic manipulations caused a substantial decrease in streptomycin resistance. The Arg 43 codon of *rpsL* gene in *Y.pestis* S19960127 is AGG. Once this codon altered to AAG would result in amino acid exchange from arginine to lysine (Fig 1), and corresponding streptomycin resistance would diminish greatly, i.e. the MIC of streptomycin was 4,096 mg/L for S19960127 (*rpsL-128G*), while the MIC decreased to 4 mg/L in strain S19960127:: *rpsL-128A* (Table 1 and Fig 2). This result indicates that *rpsL* mutation at 128 bp in the *rpsL* gene is responsible for the streptomycin resistance in strain S19960127.

## Discussion

### Streptomycin resistance worldwide and underlying mechanism

Traditional antimicrobials used for treatment and/or prophylaxis of plague patients include aminoglycosides (streptomycin and gentamicin), tetracyclines (doxycycline and tetracycline),

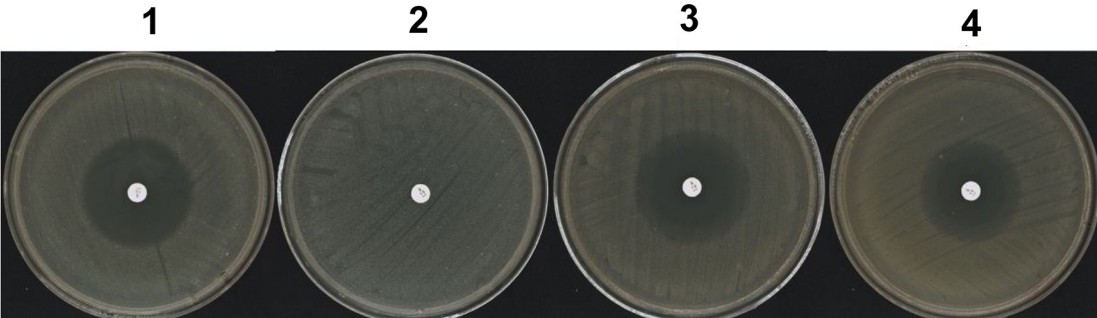

**Fig 2. Antimicrobial susceptibility test using the disk diffusion method.** *Y. pestis* strains were streaked on cation-adjusted Mueller-Hinton agar plates with streptomycin disc (300 μg/disc), incubated plates at 35°C for 48 h. 1: *Y.pestis* vaccine strain EV76 (Vaccine strain, MIC: 4 mg/L); 2: *Y.pestis* S19960127 (*rpsL-128G*, MIC 4,096 mg/L); 3: *Y.pestis* S19960127:: *rpsL-128A* (MIC 4 mg/L); 4: *Escherichia coli* ATCC 25922.

**Table 1. The Strains and Plasmids Used in this Study.**

| Strains or Plasmids | Relevant Properties | Source or Reference |
|---|---|---|
| *Y. pestis* strains | | |
| EV76 | Vaccine strain; pCD1; pPCP1; pMT1; SM<sup>S</sup> (MIC: 4mg/L), rpsL:128-A | Madagascar(1922); |
| S1996127 | Wild strain, pCD1; pPCP1; pMT1; pS96127; SM <sup>r</sup> (MIC:4,096 mg/L), *rpsL*:128-G, | This study |
| S1996127△YPO0199::*cat* | S1996127 delete YPO0199, SM <sup>r</sup>(MIC:4,096 mg/L), with cat gene | This study |
| S1996127:: *rpsL*-128A | S1996127 replacement:*rpsL*:G128-A; SM<sup>S</sup>(MIC: 4mg/L) | This study |
| *E. coli* strains | | |
| DH5α | F—, M(lacZYA-*arg*F) U169, *hsdR*17 (rk—mk <sup>+</sup>), *recA*1, *endA*1, *relA*1 | Laboratory stock |
| SM10λpir | thi recA thr leu tonA lacY supE RP4-2-Tc::Mu_::pir; Kan <sup>r</sup> | Laboratory stock |
| | SM<sup>S</sup>(MIC: 4mg/L) | |
| Plasmids | | |
| pkd3 | Cloning vector; lacZ Amp <sup>r</sup> | Laboratory stock |
| pWM91 | Suicide vector containing R6K *ori*, *sacB*, *lacZα*; Amp<sup>r</sup> | Laboratory stock |
| pWM△YPO0199::*cat* | 2.137kb, containing the flanking sequence of *rpsL*, with *cat* gene | This study |
| pWMDF1-DR2 | 1.281 kb, EV76 fragment containing the sequence of the *rpsL* | This study |

chloramphenicol, and trimethoprim/sulfamethoxazole [4]. Generally, *Y. pestis* isolates are uniformly susceptible to the dominant antibiotics against Gram-negative bacteria [27,28]. Recently, a paper reported plasmid-mediated doxycycline resistance in a *Y. pestis* strain in Madagascar (isolated from a rat in 1998) [7].

Resistance to streptomycin conferred by resistance plasmids encoding streptomycin phosphotransferase enzymes had been documented in two studies from Madagascar [5,6], in which strain 17/95 (orientalis) carried the conjugative plasmid pIP1202 (a member of the IncA/C plasmid family) and the MIC of streptomycin for strain 17/95 was >2,048 mg/L [5]. Meanwhile, another *Y. pestis* strain 16/95 (orientalis) only resistant to streptomycin, whose resistance determinant was carried by a self-transferable plasmid (pIP1203, a member of the IncP group). The MIC of streptomycin for 16/95 was 1,024 mg/L [6].

Another mechanism for streptomycin resistance is conferred *via* chromosomal mutations which alter the ribosomal binding site of streptomycin [20,26]. Streptomycin binds to the aminoacyl-tRNA recognition site (A-site) of 16S rRNA, interferes with a proofreading step in translation, and inhibits the initiation of translation, thereby perturbing polypeptide synthesis and subsequent cell death. Two gene mutations, encoding ribosomal protein S12 (*rpsL* gene) and the 16S rRNA gene (*rrs*), have been associated with streptomycin resistance in *M. tuberculosis* [29] and other species of bacteria such as *Salmonella typhimurium* [30]. Streptomycin resistance in *M. tuberculosis* is associated with substitution of amino-acid 43 or 88 in *rpsL* mutations in streptomycin-resistant isolates [26]. Highly-resistant strains are those that tolerate streptomycin concentrations >500 mg/L [20], and this phenotype has only been found in isolates with an altered *rpsL* in *M. tuberculosis*, while mutations in *rrs* are associated with a low or intermediate level of resistance (tolerating streptomycin at concentrations between 50 and 500 mg/L) in *M. tuberculosis* [20]. In this study, the high level of streptomycin resistance (the MIC was 4,096 mg/L) exhibited in *Y. pestis* S19960127 was identified as being due to a *rpsL* mutation associated with substitution of amino-acid 43 of ribosomal protein S12. Sequence comparison showed no amino-acid alteration at position 88 of ribosomal protein S12 or *rrs* gene mutations in S19960127 (Figs 1 and S1).

## Reasons for streptomycin resistance

The epidemiological investigation indicated that the Patient C took care of Patient B (primary pneumonic plague) and unfortunately got infection. Considering their clinical symptoms and treatment, as well as the genomic phylogenetic relationship in associated strains (molecular subpopulation designations see in S2 Table), we inferred that the high level of resistance by a mutation did not originate from the local reservoir of *M. himalayana* or previous plague outbreaks, because no such mutation in *rpsL* occurred in strains isolated from *M. himalayana* or human cases in adjacent counties, even though the phylogenetic lineage of these strains was same with the streptomycin-resistant strain S19960127, i.e.2.ANT2d. Such strains included S19960038 (2.ANT2d, isolated from a pneumonic plague outbreak at the end of July in 1996 in Cona County) and S19960156 (2.ANT2d, isolated from local reservoir *M. himalayana* in Cona County in 1996); In addition, S19910050 (2.ANT2d) was involved in a pneumonic plague outbreak (six-patients with five deaths) in Lhünze county in September 1991, and the strain S19910056 (2.ANT2d) was associated with a pneumonic plague outbreak (15 cases with five deaths) occurred in Qusug County (Xiajiang village, adjacent to Lhünze county) in 1991. All these strains did not exhibited substitution mutation in *rpsL* gene.

Streptomycin is the most effective antibiotic against *Y. pestis* and the drug of choice for treatment of plague, particularly for the pneumonic form[4]. The Patient C got infection from her husband (Patient B). She was given united combination treatment including streptomycin, tetracycline and trimethoprim/sulfamethoxazole for three days but still ended in death, and streptomycin resistance strain S19960127 was isolated from her necropsy organ. So the emergence of streptomycin resistance was suspected due to the streptomycin application in Patient C or her predecessor (Patient B). In addition, on one hand, the treatment failures maybe associated with such streptomycin resistance, even they were also administrated the trimethoprim/sulfamethoxazole and the tetracycline besides the streptomycin. For the former two antibiotics, comparatively, are not effective as streptomycin, especially for the treatment of pneumonic plague[4]. For the other hand, we can't completely rule out the possibility that the treatment failure was associated with the severity of pneumonic plague disease. Because no strain was obtained from Patient B, so, for Patient B, no sufficient evidence to confirm that the pathogen had developed corresponding streptomycin resistance because of streptomycin application, even though Patient B was given streptomycin treatment for 8 days. Of course, Additional studies investigating the origination of streptomycin resistance are still needed *in vitro* or *in vivo* to prove such selective function.

## Risk of streptomycin-resistant *Y. pestis* in *M. himalayana* plague focus on the Qinghai-Tibet plateau

In history, plague once killed millions of people in Europe in the 14th century and tens of thousands in China in the 19th century [9]. To date, at least 12 plague foci had been identified in China[31], in which the *Marmota himalayana* plague focus on the Qinghai-Tibet plateau is the largest and highest risk focus in China [31]. In this natural plague focus, human plague infection is always associated with hunting or skinning diseased or dead *M. himalayana* or Tibetan sheep (*Ovis aries*) [32] and the pathogen *Y. pestis* (biovar antiqua) frequently causes pneumonic or septicemic plague with high mortality [32]. So even though *Y.pestis* resistance to streptomycin is not widespread in this plague focus, we should not ignore the possibility of spread due to overuse of streptomycin, for once the streptomycin-resistant *Y. pestis* spreads in humans or in hosts and vectors in natural plague foci, a significant public health threat will have to be confronted. What's more, we can't completely rule out such a possibility that clinician attributed the treatment failure on plague to severity of the disease while ignored

streptomycin resistance in the case of the antibiotic was administrated. So, from the clinical and public health point of view, antimicrobial susceptibility monitoring of *Y. pestis* isolates should be routinely carried out in countries threatened by plague.

## The significance of this research

Streptomycin is the preferred choice for therapy of plague in China and other countries. This is the first report that streptomycin resistance is present in *Y. pestis* in China. To the best of our knowledge, this is the first report of a high level of streptomycin resistance associated with an *rpsL* mutation in *Y. pesti*s. So streptomycin susceptibility monitoring of *Y. pestis* isolates should not only include plasmid-mediated resistance but also include the ribosomal protein S12 gene (*rpsL*) mutation, especially when treatment failure is suspected due to antibiotic resistance. Currently, Genome sequencing became significantly easier with the advance of next generation sequencing technologies [23]. On one hand, the genome-wide SNP analysis could be used to illustrate the phylogenetic relationship or the microevolution of *Y. pesti*s [33], including used for source-tracking in plague outbreaks. On the other hand, whole genome sequencing and associated analysis methods could greatly facilitate the detection of known or potentially novel mutations associated with antibiotics resistance in *Y. pesti*s. Ultimately, such insights will greatly assist with patient treatment, management, and the disease control.

## Supporting information

**S1 Table. Primers used in this study.**
(DOC)

**S2 Table. The sequenced strains of *Y. pestis* using in this study.**
(XLS)

**S1 Fig. Sequence comparisons of *rrs* gene.**
(TIF)

## Acknowledgments

We thank all the professionals for natural plague foci surveillance in Shannan Prefecture and in Tibet Autonomous Region CDCs.

## Author Contributions

**Conceptualization:** Ruixia Dai, Xuefei Zhang, Jianguo Xu, Wei Li.

**Data curation:** Jian He, Yiting Wang, Youquan Xin, Juan Jin, Jixiang Bai, Wei Li.

**Formal analysis:** Jian He, Yiting Wang, Wei Li.

**Funding acquisition:** Ruixia Dai, Xuefei Zhang, Wei Li.

**Investigation:** Ruixia Dai, Jian He, Xi Zha, Xiaoyan Yang, Youquan Xin, Sheng Li, Juan Jin, Qi Zhang, Jixiang Bai, Yao Peng, Baiqing Wei.

**Methodology:** Jian He, He Gao, Juan Li, Wei Li.

**Project administration:** Ruixia Dai, Wei Li.

**Resources:** Xi Zha, Xuefei Zhang, Juan Li, Hailian Wu, Qingwen Zhang.

**Software:** Yiting Wang.

**Supervision:** Ruixia Dai, Jianguo Xu, Wei Li.

**Validation:** Xi Zha, Baiqing Wei, Wei Li.

**Visualization:** Jian He, Yiting Wang, Yumeng Wang.

**Writing – original draft:** Ruixia Dai, Xi Zha, Wei Li.

**Writing – review & editing:** Wei Li.

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
