## [Decision Letter · Decision Letter 0]

7 Jan 2021

Dear Dr. Li,

Thank you very much for submitting your manuscript "A Novel Mechanism of Streptomycin Resistance in Yersinia pestis: Mutation in the rpsL gene" for consideration at PLOS Neglected Tropical Diseases. As with all papers reviewed by the journal, your manuscript was reviewed by members of the editorial board and by several independent reviewers. The reviewers appreciated the attention to an important topic. Based on the reviews, we are likely to accept this manuscript for publication, providing that you modify the manuscript according to the review recommendations. 

This novel mechanism of Streptomycin resistance in Yersinia pestis is of interest for Public Health and deserves to be published. However the reviewers requested a series of precisions to clarify the experiments of gene replacement and some minor corrections.

Comments from the reviewer 1 are listed in an attached document.

Sincerely,

Anne-Sophie Le Guern

Guest Editor

Javier Pizarro-Cerda

Deputy Editor

Thank you very much for submitting your manuscript. 

This novel mechanism of Streptomycin resistance in Yersinia pestis is of interest for Public Health and deserves to be published. However the reviewers requested a series of precisions to clarify the experiments of gene replacement and some minor corrections.

Comments from the reviewer 1 are listed in an attached document.

Reviewer's Responses to Questions

**Key Review Criteria Required for Acceptance?**

**Methods**

-Are the objectives of the study clearly articulated with a clear testable hypothesis stated?

-Is the study design appropriate to address the stated objectives?

-Is the population clearly described and appropriate for the hypothesis being tested?

-Is the sample size sufficient to ensure adequate power to address the hypothesis being tested?

-Were correct statistical analysis used to support conclusions?

-Are there concerns about ethical or regulatory requirements being met?

Reviewer #1: (No Response)

Reviewer #2: The methods appear sound.

Reviewer #3: The objectives of the study are clearly formulated with a clearly formulated testable hypothesis that the resistance of the clinical isolate of Yersinia pestis to streptomycin is determined by changes in its genome that occurred during the infectious / epidemic process during treatment with streptomycin.

The study design is appropriate to address the stated objectives.

The next questions are not applicable.

**Results**

-Does the analysis presented match the analysis plan?

-Are the results clearly and completely presented?

-Are the figures (Tables, Images) of sufficient quality for clarity?

Reviewer #1: (No Response)

Reviewer #2: The results seem fine and support the conclusion that the K43R mutation in the S12 protein confers streptomycin resistance to Y. pestis.

Reviewer #3: The analysis presented match the analysis plan.

The results are clearly and completely presented.

The Tables & Figures are of sufficient quality for clarity.

**Conclusions**

-Are the conclusions supported by the data presented?

-Are the limitations of analysis clearly described?

-Do the authors discuss how these data can be helpful to advance our understanding of the topic under study?

-Is public health relevance addressed?

Reviewer #1: (No Response)

Reviewer #2: The conclusions are reasonable.

Reviewer #3: Yes.

Not applicable.

Yes.

Yes.

**Editorial and Data Presentation Modifications?**

Reviewer #1: (No Response)

Reviewer #2: None required.

Reviewer #3: Lines 47-48: “… ANOTHER NEW BIOVAR, INTERMEDIUM, was proposed according to genetic diversity research [39]. Therefore, Y. pestis can be assigned into FIVE BIOVARS—Antiqua (glycerol positive, arabinose positive, and nitrate positive), Medievalis (glycerol positive, arabinose positive, and nitrate negative), Orientalis (glycerol negative, arabinose positive, and nitrate positive), Microtus (glycerol positive, arabinose negative, and nitrate negative), and Intermedium (its biochemical features still need to be determined) [38]”.

“… pestis [39]. From the MLVA analysis, A NEW BIOVAR, INTERMEDIUM, was proposed to describe rhamnose-positive Y. pestis subsp. pestis strains that occasionally infect humans and that are isolated mostly from marmots in the northern Tian Shan Mountains in China [39] … 

” [Qi Z, Cui Y, Zhang Q, Yang R. Taxonomy of Yersinia pestis. Adv Exp Med Biol. 2016;918:35-78. doi: 10.1007/978-94-024-0890-4_3].

Minor improvement in English required.

Restriction enzyme names have not been written in italics for a long time.

Line 328: Serovar names do not need to be written in italics.

Line 340: Is Table S1 really needed?

**Summary and General Comments**

Reviewer #1: (No Response)

Reviewer #2: Dai et al. present a paper demonstrating a mutation in the rpsL gene of Y. pestis that confers high-level resistance to streptomycin. The paper as such is well written and the results convincingly demonstrate that the K43R mutation leads to streptomycin resistance. This apparently the first case of a chromosomal mutation reported for Y. pestis that increases resistance to streptomycin.This is of interest and medical significance. 

However, I am not sure that the study goes far enough to warrant publication in PLoS Neglected Tropical Diseases. As the authors note in Figure 1, the same mutation is known to confer resistance in Enterobacteriaceae. To increase the impact of their study, the authors could conduct an evolution experiment with increasing to concentrations of streptomycin to find out how easily the rpsL mutation arises. In addition, the authors should find out whether the mutation has any fitness costs by measuring growth curves or possibly performing in vivo competition assays against the wild-type version of the gene. If there are fitness costs, does the genome contain any potential compensatory mutations? What effects might these have?

Reviewer #3: A team of Chinese scientists have prepared materials for publication that are especially relevant during the current pandemic. Their work once again reminds us that there are temporary difficulties and the problems that have accompanied man since prehistoric times. Adequate attention to these problems will save humanity from troubles that will make the current pandemic seem like a child's play. Overall, the manuscript is well written, but requires minor editing in English.

PLOS authors have the option to publish the peer review history of their article (what does this mean?). If published, this will include your full peer review and any attached files.

Reviewer #1: No

Reviewer #2: No

Reviewer #3: Yes: Andrey P. Anisimov
---

## [Decision Letter · Decision Letter 1]

23 Mar 2021

Dear Dr. Li,

We are pleased to inform you that your manuscript 'A Novel Mechanism of Streptomycin Resistance in Yersinia pestis: Mutation in the rpsL gene' has been provisionally accepted for publication in PLOS Neglected Tropical Diseases.

Best regards,

Anne-Sophie Le Guern

Guest Editor

Javier Pizarro-Cerda

Deputy Editor

Reviewer's Responses to Questions

**Key Review Criteria Required for Acceptance?**

**Methods**

-Are the objectives of the study clearly articulated with a clear testable hypothesis stated?

-Is the study design appropriate to address the stated objectives?

-Is the population clearly described and appropriate for the hypothesis being tested?

-Is the sample size sufficient to ensure adequate power to address the hypothesis being tested?

-Were correct statistical analysis used to support conclusions?

-Are there concerns about ethical or regulatory requirements being met?

Reviewer #1: All the methods seem appropriate and are clearly formulated.

Could you reintroduce the Supplemental Table 1 from the first version of the manuscript as it disappeared in this revised version. Indeed it is important to have the details of the Y. pestis used for antibiotic resistance evaluation in this study. Please reannotate subsequent numbers of supplemental Table then.

**Results**

-Does the analysis presented match the analysis plan?

-Are the results clearly and completely presented?

-Are the figures (Tables, Images) of sufficient quality for clarity?

Reviewer #1: Results presented here match the analysis plan and support the main result stating that K43R mutation within S12 protein is associated with high-level resistance of Y. pestis to streptomycin.

**Conclusions**

-Are the conclusions supported by the data presented?

-Are the limitations of analysis clearly described?

-Do the authors discuss how these data can be helpful to advance our understanding of the topic under study?

-Is public health relevance addressed?

Reviewer #1: The conclusions are well supported by the data and results.

Public health relevance is addressed as it highlights for the potential threaten of this mutation spread within Y. pestis population. Furthermore, this study points the need for the surveillance not only restricted to the acquisition of strA and strB in case of streptomycin resistance of Y. pestis strains.

**Editorial and Data Presentation Modifications?**

Reviewer #1: (No Response)

**Summary and General Comments**

Reviewer #1: This revised version of the manuscript entitled “A Novel Mechanism of Streptomycin Resistance in Yersinia pestis: Mutation in the rpsL gene” by Ruixia Dai et al. reports a non-described so far in Yersinia mutation in the rpsL gene associated with Streptomycin resistance.

In the lights of previous reviewers's comments it significantly improved its clarity and meaningful of the results.

PLOS authors have the option to publish the peer review history of their article (what does this mean?). If published, this will include your full peer review and any attached files.

Reviewer #1: No

---

## [Editor Report · Acceptance letter]

16 Apr 2021

Dear Dr. Li,

We are delighted to inform you that your manuscript, "A Novel Mechanism of Streptomycin Resistance in Yersinia pestis: Mutation in the rpsL gene," has been formally accepted for publication in PLOS Neglected Tropical Diseases.

Best regards,

Shaden Kamhawi

co-Editor-in-Chief

Paul Brindley

co-Editor-in-Chief
